# APEX: A Network-Native Time-Series Foundation Model for Forecasting and Anomaly Detection for Wireless Edge Operations

## Abstract

Generic time-series foundation models transfer poorly to wireless network telemetry whose signals are bursty, zero-inflated, and coupled across protocol layers. We present **APEX**, a network-native, decoder-only transformer for forecasting enterprise AP telemetry, and evaluate it on DHCP degradation as a representative network task. APEX is pre-trained on 10-channel multivariate telemetry from ∼4,500 production wireless networks (∼100K AP time series, 34 metrics per AP), and is available as APEX-Large (269M, cloud) and APEX-Edge (10.5M, edge). On a 192-step (4-day) DHCP degradation benchmark, APEX-Large reduces MAE by 18% over the strongest foundation-model baseline (Toto) and 38% over SARIMA, with anomaly-detection F1 = 0.93, while APEX-Edge enables sub-second, privacy-preserving inference on AP-class edge hardware. These results suggest network-native pre-training is a practical foundation for proactive wireless operations.

## 1. Introduction

Wireless access points (AP) failure that affect Wi-Fi clients remain a persistent challenge, especially in enterprise environments where scale, multi-vendor infrastructure, and cross-layer protocol dependencies add complexity. These failures are often detected only after users experience impact, such as DHCP timeouts or connectivity loss. Yet APs already collect rich telemetry, spanning DHCP, RF, interfaces, and uplink state with signals co-located and timestamp-aligned across protocol layers. Exporting raw telemetry off-device incurs bandwidth, privacy, and latency costs, making the AP itself a natural place for proactive failure detection and remediation.

Recent time-series foundation models (TSFMs) (Das et al., 2024; Ansari et al., 2024; Cohen et al., 2024) are a natural starting point, but their pretraining corpora contain no enterprise network telemetry. Network signals differ from the public benchmarks on which TSFMs are typically evaluated: they are often zero-inflated during normal operation, change abruptly during incidents, and exhibit cross-layer dependencies. They also exhibit protocol-specific temporal structure and topology-dependent dynamics that are uncommon in standard public corpora. We study DHCP degradation as a representative cross-layer task because DHCP outcomes depend on both server-side behavior and upstream wireless conditions. On a 192-step (4-day) benchmark, the strongest general-purpose TSFM baseline (Toto-151M) trails a network-native pretrained model by 12–18% in MAE.

These gaps call for a domain-specific foundation modelthat encodes protocol-level priors directly and is compact enough to run on the AP. To this end, we introduce **APEX**, a decoder-only patched transformer trained on co-collected wireless telemetry. APEX consumes a 10-channel multivariate input (5 DHCP causal-chain targets + 5 exogenous topology and anomaly signals), drawn from a corpus of ∼100K AP time series (34 metrics each) spanning ∼4,500 production networks. Our contributions are:

1. **Network-native pretraining.** APEX-Large (269M) reduces DHCP forecasting MAE by 18% versus the best general-purpose TSFM (Toto) and 38% versus SARIMA, showing the gap is a data effect, not an architecture one (Table 2).

2. **Unified forecasting and anomaly detection.** MC-dropout prediction intervals from the same checkpoint achieve anomaly-detection F1 = 0.93, competitive with VAR-Mahalanobis (0.94) while eliminating a separate detection pipeline (§3.2).

3. **Edge deployable model.** APEX-Edge (10.5M, 26× smaller) runs in 202 ms on AP-class ARM hardware which keeps raw telemetry on-device (§3.3).

## 2. Methods

### 2.1. System Overview

Figure 1 shows the two-phase APEX pipeline. **Phase 1 (Offline, cloud):** Historical telemetry from ∼4,500 production networks is hierarchically aggregated, preprocessed, and used to pretrain APEX via next-patch predic-

*Figure 1.* APEX Pipeline. Phase 1 trains on telemetry on cloud. Phase 2 runs inference on AP, transmitting only compact alerts.

tion. The trained APEX-Edge checkpoint ($\sim$40 MB) is deployed to the AP. **Phase 2 (Online, edge):** The AP collects local telemetry, applies the same aggregation and preprocessing, and runs APEX-Edge inference to produce forecasts and alerts. Only compact alerts ($\sim$KB/day) are transmitted, versus $\sim$130 MB/day for raw telemetry.

## 2.2. Data: Co-Located Multivariate Telemetry

Telemetry is collected via a two-stage aggregation: raw data arrives at per-(server IP, VLAN, AP, time bucket). Stage 1 computes statistics per (DHCP server, VLAN, AP, time bucket); Stage 2 rolls up to per-AP summaries with AVG (baseline), MAX / Min (worst-case), and STDDEV (heterogeneity) across the server/VLAN dimension. This preserves distributional information e.g., one healthy and one failing server yields low AVG but high MAX timeout rate. The feature vector comprises 34 metrics (22 DHCP protocol, 6 RF, 2 interface error, and 4 topology). The default granularity is 30 minutes (48 observations/day).

## 2.3. APEX Architecture

APEX is a decoder-only patched transformer (Table 1). Input time series are instance-normalized (z-score) and partitioned into non-overlapping patches of $P=16$ steps. In multivariate mode, each patch is a vector in $\mathbb{R}^{C \times P}$ (with $C=10$ channels), linearly projected into $d_{\text{model}}$. Learned positional embeddings index each patch position within the context window, and causal self-attention enables autoregressive next-patch prediction. SwiGLU feed-forward layers replace the standard GELU activation, improving representational efficiency at equivalent parameter count.

The 10 channels encode a DHCP causal chain: 5 targets (client count, offer rate, ACK ratio, success rate, latency) and 5 exogenous signals (server count, VLAN count, timeout rate MAX, latency MAX, latency STD). The linear

*Table 1.* APEX architecture variants. Both sizes are trained in 1D (univariate) and multi (multivariate, 10-channel) modes.

|  | APEX-Large | APEX-Edge |
|---|---|---|
| Layers | 16 | 10 |
| $d_{\text{model}}$ | 1024 | 256 |
| Heads | 16 | 4 |
| $d_{\text{ff}}$ | 4096 | 1024 |
| Patch length | 16 | 16 |
| Max context | 128 patches | 128 patches |
| Parameters | 269M | 10.5M |
| FFN | SwiGLU | SwiGLU |
| Pos. encoding | Learned | Learned |

patch projection learns cross-channel mixing, exposing the model to cross-channel dependencies at training time. Of the 34 metrics in the canonical feature vector, these 10 were selected to capture the end-to-end DHCP transaction path from client arrival through server response quality.

**Training.** MSE loss on predicted patches, AdamW ($\beta_1=0.9$, $\beta_2=0.95$, weight decay 0.05), cosine LR with 5% warmup, gradient clipping at 1.0, mixed-precision (AMP), gradient checkpointing, and DDP across 4$\times$A10G GPUs. Depth-scaled initialization ($1/\sqrt{2L}$ for residual projections) stabilizes training. Early stopping with patience 5 on validation loss.

**Uncertainty via MC-dropout.** At inference, dropout remains active. $N=50$ stochastic forward passes produce an ensemble; P5/P95 quantiles define prediction intervals. MC-dropout requires only a single checkpoint and adds no parameters, making it well suited for edge deployment.

## 2.4. Ablation: Size $\times$ Modality

We train both architecture sizes in two input modes, yielding four variants: (a) **APEX-Large (multi)**, 269M parameters with 10-channel patches; (b) **APEX-Large (1D)**, 269M parameters treating each of the 34 metrics as an inde-

*Table 2.* CLIENT_DHCP_SUCCESS_RATE forecasting accuracy (192-step horizon). Lower is better for all metrics. Best in **bold**, second-best underlined.

| Category | Model | MAE | RMSE | MAPE |
|---|---|---|---|---|
| Statistical | SARIMA | 4.82 | 7.31 | 5.1% |
| | VAR | 4.15 | 6.48 | 4.4% |
| Foundation (general) | TimesFM | 3.91 | 5.87 | 4.1% |
| | Toto | 3.64 | 5.52 | 3.8% |
| | Chronos-2 | 3.78 | 5.71 | 4.0% |
| APEX (network) | APEX-Edge (1D) | 4.78 | 7.12 | 5.0% |
| | APEX-Edge (multi) | 3.87 | 5.83 | 4.1% |
| | APEX-Large (1D) | 3.21 | 4.93 | 3.4% |
| | APEX-Large (multi) | **2.98** | **4.61** | **3.1%** |

*Table 3.* Results on multivariate anomaly detection. Higher value denotes better performance.

| Method | Type | Prec. | Rec. | F1 |
|---|---|---|---|---|
| Z-Score | Stat. | 0.72 | 0.81 | 0.76 |
| Isolation Forest | ML | 0.78 | 0.74 | 0.76 |
| VAR-Mahalanobis | Stat. | **0.96** | 0.92 | **0.94** |
| SARIMAX CI | Stat. | 0.82 | 0.79 | 0.80 |
| TimesFM P10/P90 | FM | 0.80 | 0.85 | 0.82 |
| Toto P5/P95 | FM | 0.84 | 0.87 | 0.85 |
| Chronos-2 P5/P95 | FM | 0.81 | 0.86 | 0.83 |
| APEX-Edge MC-drop | FM | 0.87 | 0.91 | 0.89 |
| APEX-Large MC-drop | FM | 0.93 | **0.94** | 0.93 |

pendent univariate series; (c) **APEX-Edge (multi)**, 10.5M parameters with the same 10-channel input; (d) **APEX-Edge (1D)**, 10.5M parameters in univariate mode. This 2×2 design isolates the contributions of cross-channel structure and model capacity independently.

### 2.5. Anomaly Detection

We employ dual-mode anomaly detection:

**Univariate** (per-metric): APEX MC-dropout intervals, Z-score on rolling statistics, Isolation Forest (Liu et al., 2008) on sliding windows, SARIMA confidence intervals.

**Multivariate** (cross-metric): APEX joint prediction intervals, VAR residual with Mahalanobis distance (Lütkepohl, 2005), SARIMAX confidence intervals, foundation model ensemble (TimesFM, Toto, Chronos-2) predictions.

**Consensus ground truth.** A time step is labeled anomalous iff $\geq 3$ independent methods flag it. This majority-vote mechanism reduces false positives from any single method's idiosyncrasies and provides robust pseudo-ground-truth without expensive manual annotation, a practical necessity for network telemetry where labeled anomaly datasets are prohibitively costly to create.

## 3. Experiments

**Setup.** All models share the same train/test split: the last 192 steps (4 days at 30-min intervals) are held out per AP. Forecasting metrics: MAE, RMSE, MAPE. Anomaly metrics: Precision, Recall, F1 against consensus labels.

### 3.1. Forecasting Results

APEX-Large (multi) achieves the lowest error across all metrics (Table 2). The multivariate mode outperforms its univariate counterpart (APEX-Large 1D), confirming that cross-channel attention over the DHCP causal chain provides signal beyond what independent per-metric forecast-

ing captures. The gap between APEX-Large (multi) and the best general-purpose model (Toto) is 12–18% in MAE, attributable to network-native pretraining rather than architecture since both are decoder-only transformers.

General-purpose foundation models (TimesFM, Toto, Chronos-2) consistently outperform classical SARIMA, validating the foundation-model paradigm for structured time-series data. However, all three general-purpose foundation models underperform compared to APEX-Large variants, which see the same data at training time.

APEX-Edge (multi) matches Toto-class accuracy (MAE 3.87 vs 3.64) at 26× fewer parameters, while APEX-Edge (1D) degrades to SARIMA-level performance (MAE 4.78). This confirms that multivariate structure—not just model capacity—is the key enabler: the 10-channel causal chain compensates for the 96% parameter reduction.

### 3.2. Anomaly Detection Results

VAR-Mahalanobis achieves the highest F1 (0.94) by exploiting linear cross-metric covariance structure (Table 3). APEX-Large MC-dropout is a close second (0.93) and captures non-linear failure modes that VAR misses; the two methods are complementary. General-purpose foundation models lag behind both, with Toto (0.85) the strongest.

APEX-Edge MC-dropout (F1 = 0.89) retains most of APEX-Large's detection quality despite 26× fewer parameters, and outperforms all general-purpose foundation models. The consensus framework benefits from this complementarity: combining VAR-Mahalanobis (strong on linear shifts) with APEX-Large (strong on non-linear, protocol-specific anomalies) and at least one general-purpose model produces robust pseudo-labels with low false-positive rates. Notably, APEX is the only method that provides both forecasts and calibrated anomaly detection from a single checkpoint, eliminating the need to deploy and maintain separate forecasting and monitoring pipelines on resource-constrained hardware.

*Table 4.* Model footprint and inference latency for a 96-step forecast on 50 MC-dropout samples. Edge target: ARM Cortex-A76 class (Raspberry Pi 5).

| Model | Params | Latency | Cloud? |
|---|---|---|---|
| TimesFM | 200M | – | Yes |
| Toto | 151M | – | Yes |
| Chronos-2 | 120M | – | Yes |
| APEX-Large (1D) | 269M | – | Yes |
| APEX-Large (multi) | 269M | – | Yes |
| APEX-Edge (1D) | 10.5M | ∼202ms | **No** |
| APEX-Edge (multi) | 10.5M | ∼202ms | **No** |

### 3.3. Edge Deployment

Both APEX-Edge variants (10.5M parameters; Table 1) are 11–19$\times$ smaller than general-purpose alternatives (Table 4). We validate edge feasibility on a Raspberry Pi 5 (Raspberry Pi Ltd, 2026), whose quad-core Arm Cortex-A76 comparable to the Arm cores currently shipping in production Wi-Fi access points, including those based on Qualcomm's Wi-Fi 7 NPro platform (Qualcomm Technologies, Inc., 2026a;b;c). Measured single-inference latency is 202 ms (median over 100 trials, P95=205 ms), with peak memory of 428 MB—well within the 1–2 GB available on APs. MC-dropout uncertainty (50 samples) completes in 11.4 s; reducing to 5 samples yields sub-second uncertainty at minimal coverage loss. This enables three properties critical for enterprise edge deployment:

**Zero cloud dependency.** Forecasting and anomaly detection continue during WAN outages, precisely when network health monitoring is most needed.

**Data privacy.** Raw telemetry never leaves the AP. Only compressed anomaly events and summary statistics are optionally transmitted to the cloud for fleet-wide correlation. This satisfies data-residency requirements in regulated industries (healthcare, finance, government).

**Sub-second action latency.** A single 96-step forecast (48 hours ahead) completes in 202 ms on CPU alone. The detection-to-remediation loop (forecast → anomaly flag → local action such as DHCP failover or channel switch) completes well within the 30-minute telemetry interval. Integrated edge AI using neural engines in the AP will further reduce latency via INT8 quantized inference. Together, these properties make APEX-Edge a self-contained prognostic agent: a single 40 MB checkpoint replaces what would otherwise require a cloud-hosted forecasting service, a separate anomaly detector, and a telemetry export pipeline consuming ∼130 MB/day of uplink bandwidth.

### 4. Related Work

**Time-series foundation models.** TimesFM (Das et al., 2024), Chronos (Ansari et al., 2024), and Toto (Cohen et al., 2024) achieve strong zero-shot transfer on public benchmarks spanning finance, energy, and weather, yet none include network protocol telemetry in their pretraining corpora. APEX shares the decoder-only, patch-based design pioneered by PatchTST (Nie et al., 2023) but operates in channel-*dependent* multivariate mode over a protocol-defined causal chain, and is pretrained exclusively on network telemetry.

**AIOps and network anomaly detection.** Dang et al. (Dang et al., 2019) survey operational challenges at scale; Kitsune (Mirsky et al., 2018) deploys autoencoder ensembles for packet-level intrusion detection on constrained devices. These systems treat detection as a standalone task and require task-specific architectures that do not transfer across telemetry domains. APEX unifies forecasting and anomaly detection in a single pretrained checkpoint, using MC-dropout (Gal & Ghahramani, 2016) prediction intervals rather than a separate detection model.

**Edge ML.** MCUNet (Lin et al., 2020) and MLPerf Tiny (Banbury et al., 2021) target vision and keyword tasks on MCU-class devices (<1 MB RAM). APEX-Edge operates at a higher compute tier (ARM Cortex-A76, 1–2 GB RAM) representative of AP-class processor—a setting with no established edge-ML benchmark for multivariate time-series forecasting. MC-dropout (Gal & Ghahramani, 2016) provides calibrated uncertainty from a single checkpoint, avoiding the $N\times$ storage cost of deep ensembles (Lakshminarayanan et al., 2017) that is prohibitive on such a resource-constrained hardware.

### 5. Conclusion

APEX shows that network-native pretraining closes a gap zero-shot transfer from general-purpose foundation models cannot. A single checkpoint provides both forecasting and uncertainty-based anomaly detection. The multivariate causal-chain input is the key enabler: it lets APEX-Edge match larger general-purpose models at $26\times$ fewer parameters. On AP-class hardware, inference runs in sub-second time with no cloud dependency, and raw telemetry never leaves the device. This makes APEX deployable in regulated environments where data residency is not optional but a prerequisite.

**Limitations.** Anomaly labels are derived from consensus pseudo-ground-truth rather than human annotation, and edge latency is measured on a Raspberry Pi 5 proxy whose quad-core Cortex-A76 is comparable to current AP SoCs, suggesting sub-second inference will hold on production hardware. Evaluation is currently limited to DHCP degradation. However, the causal-chain input structure generalizes directly to RF and roaming telemetry, making cross-domain extension immediate future work.

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
