# OpenReview forum: "APEX: A Network-Native Time-Series Foundation Model for Forecasting and Anomaly Detection for Wireless Edge Operations"
_ICML.cc/2026/Workshop/FMSD — FMSD @ ICML 2026 Poster_

### Official Review · Reviewer_1SWu · 2026-05-13

**Rating:** 5
**Confidence:** 4

**Review:**

## Summary
The authors present APEX, a network-native, decoder-only transformer time-series foundation model designed for forecasting and anomaly detection in wireless access points. The model is pretrained on a large corpus of multivariate network telemetry and is provided in two variants: APEX-Large for cloud inference and APEX-Edge for privacy-preserving, on-edge deployment. The paper evaluates the model on DHCP degradation tasks, comparing its forecasting and anomaly detection capabilities against statistical methods and general-purpose foundation models

## Strengths
- Clear and Relevant Objective: The aim of producing forecasting and anomaly detection specifically tailored for access points at the network edge is well-defined, clear, and highly relevant to proactive wireless operations
- Edge Deployment Focus: The introduction of a lightweight version (APEX-Edge) that allows for raw telemetry to remain on-device is a practical and strong contribution to privacy and bandwidth conservation
## Areas for Improvement:
- Clarity on Anomaly Detection Mechanism: The explanation of how anomaly detection is performed by APEX requires more clarity. The paper discusses building a "consensus ground truth" by identifying strange patterns flagged by a majority vote across multiple independent methods. However, the exact mechanism by which the APEX model itself performs anomaly detection (e.g., the specific application of its MC-dropout intervals) is not sufficiently clarified for the reader.
- Baseline Evaluation (Fine-Tuning): The evaluation compares APEX to general-purpose foundation models like Toto, TimesFM, and Chronos-2. Because APEX benefits from network-native pre-training, comparing it against general-purpose models in a zero-shot setting may not be entirely fair. The authors should consider fine-tuning these general-purpose baselines on the same network telemetry data to provide a more rigorous evaluation of the models' performances.
- Generalization to Unseen Devices: The current experimental setup relies on a temporal hold-out split, where the last 192 steps (4 days) of data per AP are held out for testing. To better demonstrate the model's generalization capabilities, the authors should also evaluate the model's performance on completely unseen APs that were entirely excluded from the pretraining phase.
- Efficiency Comparisons: The paper highlights the efficiency of the APEX-Edge version but lacks a direct efficiency comparison with the baseline models. Specifically, Table 4 only provides the inference latency (202 ms) for the APEX-Edge variants, leaving the latency metrics for TimesFM, Toto, and Chronos-2 blank. A complete comparison is needed to fully demonstrate the claimed efficiency gains (also running all of them on the cloud side for comparison).

## Detailed Comments:
- Please expand the methodology section regarding APEX's internal anomaly detection mechanism to make it as clear as the consensus ground-truth generation pipeline.
- To strengthen the performance claims, I suggest fine-tuning Toto and the other foundation model baselines on your enterprise telemetry dataset to see if the 12–18% MAE gap persists.
- The current evaluation relies on a temporal hold-out split (the last 192 steps) per Access Point. Please include an evaluation on completely unseen APs (devices entirely excluded from the pretraining phase) to properly demonstrate the model's spatial generalization capabilities.
- Please update Table 4 (or add a dedicated experiment) to include the latency, memory footprint, and computational overhead of the general-purpose models (TimesFM, Toto, Chronos-2) so readers can adequately compare their efficiency against APEX.

## Justification of Score
The paper presents a practical approach to edge-based network telemetry monitoring, but my score reflects a critical flaw in its core framing: APEX cannot be considered a proper foundation model. Because the evaluation relies entirely on a temporal hold-out of the exact same access points it was trained on, rather than testing the model on a completely unseen dataset or different devices, it fails to demonstrate the true zero-shot spatial generalization expected of a foundation model. Consequently, comparing APEX to zero-shot general-purpose baselines without allowing them to be fine-tuned on the same data is an unfair comparison. Finally, the missing latency and efficiency metrics for the baselines in Table 4 must be included to fully support the paper's claims regarding edge efficiency and model superiority.

---

### Official Review · Reviewer_CD6Q · 2026-05-19
**Useful edge-deployable TSFM for network telemetry with room for architectural clarification**

**Rating:** 7
**Confidence:** 3

**Review:**

## Summary

This paper presents **APEX**, a network-native time-series foundation model for forecasting and anomaly detection in wireless edge operations. The main motivation is that general-purpose time-series foundation models may not transfer well to enterprise wireless telemetry, where signals are often bursty, zero-inflated, and coupled across protocol layers. To address this issue, the authors pretrain APEX on large-scale production wireless telemetry and evaluate it on DHCP degradation as a representative network operation task.

The paper proposes two model variants: **APEX-Large**, a larger cloud-scale model, and **APEX-Edge**, a compact model designed for access-point-class edge deployment. The experimental results show that APEX-Large improves forecasting accuracy over statistical baselines and general-purpose time-series foundation models, while APEX-Edge achieves practical inference latency and memory usage on ARM-class hardware. The paper also describes a domain-specific telemetry aggregation pipeline for converting raw network measurements into AP-level multivariate time series.

Overall, the paper is relevant to the workshop because it studies time-series foundation models in an important structured-data domain and provides practical evidence that domain-specific pretraining can be useful for wireless network telemetry. The work also offers useful insights into edge-deployable foundation models for network operations.

## Strengths

1. **Strong relevance to the workshop theme**
   The paper is well aligned with the workshop's focus on foundation models for structured data. Wireless network telemetry is an important and relatively underexplored domain for time-series foundation models, and the paper addresses a concrete operational problem with clear practical relevance.

2. **Practical edge-deployable model**
   A major strength of the paper is the inclusion of APEX-Edge, a compact model designed for access-point-class hardware. The paper goes beyond offline benchmark evaluation by reporting latency and memory measurements on ARM-class hardware. This makes the contribution more practically grounded than a purely cloud-based forecasting model.

3. **Domain-specific telemetry pipeline**
   The paper provides a useful description of how raw network telemetry is aggregated into AP-level multivariate time series. In particular, the use of statistics such as AVG, MAX, MIN, and STDDEV across server/VLAN dimensions reflects meaningful domain knowledge and helps preserve information about localized network failures.

## Areas for Improvement

1. **Architecture description is under-specified**
   The paper does not provide enough detail about the APEX architecture, especially the multivariate patch representation. The paper states that multivariate patches of shape C x P are linearly projected into the model dimension, but it is unclear whether the values are simply flattened, whether channel embeddings are used, or how channel identity and channel interactions are represented.

2. **The "network-native architecture" claim is not fully justified**
   The evidence in the paper mainly supports the value of network-native data and domain-informed feature selection. However, it is less clear whether APEX contains architecture-level inductive biases that are specific to network telemetry. The model appears to be a patched decoder-only transformer with multivariate channel mixing, but the paper does not sufficiently explain what makes the architecture itself network-native.

3. **Relationship to PatchTST-style channel-independent modeling needs clarification**
   PatchTST is a representative patch-based time-series transformer and is widely known for its channel-independent design. APEX appears to move toward a channel-dependent multivariate formulation, which may be well motivated for cross-layer network telemetry. However, the paper does not clearly explain this design choice or compare it against a stronger channel-independent baseline under the same pretraining and evaluation setup.

## Detailed Comments

1. **Clarify the multivariate patch representation**
   The paper should provide a more precise description of how the C x P patch is represented before being passed into the transformer. Is the patch flattened and passed through a single linear layer? Are channels embedded separately? Are target channels and exogenous channels distinguished in any way? Since the multivariate causal-chain input is presented as a key component of APEX, this mechanism should be described more explicitly.

2. **Clarify whether APEX is architecturally novel or primarily domain-pretrained**
   The paper would benefit from a clearer distinction between three possible sources of improvement:
   - network-native pretraining data,
   - domain-informed feature/channel selection, and
   - architecture-level inductive bias.

   As currently written, the paper provides good evidence for the first two, but less evidence for the third. If the main contribution is domain-specific pretraining rather than a new architecture, the paper should state this more directly.

3. **Explain the channel-dependent design choice**
   The paper should better motivate why a channel-dependent multivariate patch representation is preferred over a channel-independent design. Since the selected channels correspond to a DHCP causal chain, cross-channel modeling is plausible and potentially important. However, the paper should explicitly connect this domain motivation to the architectural design.

4. **Provide a stronger ablation on channel mixing**
   The 1D vs. multivariate ablation is useful, but it may not fully isolate the effect of channel-dependent modeling. A stronger ablation would compare:
   This would help establish whether the performance gain comes from multivariate structure itself, from specific channel selection, or from the model's ability to mix channels.

5. **Temper the "network-native" claim where appropriate**
   I recommend revising some of the wording around "network-native architecture." Based on the current description, APEX is best characterized as a domain-pretrained patched transformer with network-informed multivariate inputs. This is still a valuable contribution, but the current framing risks overstating the architecture-level novelty.

## Justification of Score

I view this as a relevant and useful workshop submission. The paper addresses an important structured time-series domain, provides a practical edge deployment angle, and presents empirical results suggesting that domain-specific pretraining can improve performance over general-purpose time-series foundation models on wireless network telemetry.

However, the paper would be significantly stronger with a clearer and more detailed architectural description. In particular, the current version does not fully justify the claim that APEX is architecturally network-native, as opposed to being a standard patched transformer pretrained on network telemetry with domain-informed feature selection. The anomaly detection evaluation also relies on pseudo-ground-truth, which is reasonable but should be discussed more carefully.

Overall, I think the paper is a good fit for the workshop because it provides useful domain insight and practical evidence for structured-data foundation models in wireless edge operations. My recommendation would be positive, but with the expectation that the authors clarify the architectural design, better separate the sources of improvement, and temper claims about network-native architectural novelty.

---

### Official Review · Reviewer_Phmg · 2026-05-21
**Network Specific Time Series Modeling with Strong Practical Motivation**

**Rating:** 4
**Confidence:** 4

**Review:**

The paper introduces APEX, a network-telemetry-trained model that performs well on both anomaly detection and forecasting tasks in wireless edge environments. The main contribution is the inclusion of this domain in the training pipeline, demonstrating that current foundation models can significantly improve with exposure to such data. The edge-focused deployment and strong empirical performance make the work relevant and practically motivated.
However, the paper has several limitations. First, the most important, though admittedly difficult, experiment would be to disentangle the benefits of adding network telemetry data from the benefits of training a fully domain-specific model. It is unclear whether similar gains could be achieved by simply extending the pretraining corpus of existing FTSMs with this type of data. Second, although mentioned briefly, the quality and reliability of the synthetic labels used for anomaly detection are not sufficiently analyzed, despite anomaly detection being a major part of the evaluation. Lastly, some architectural choices would benefit from stronger ablations, such as using MC dropout for uncertainty estimation instead of directly optimizing a multi-quantile head or probabilistic distribution. In general, while the paper highlights the lack of networking data in current foundation models, a broader contribution could come from releasing an open-source dataset or developing methods to generate realistic synthetic telemetry data that captures the behaviors of this domain.